# Synthetic ShK-like Peptide from the Jellyfish *Nemopilema nomurai* Has Human Voltage-Gated Potassium-Channel-Blocking Activity

**DOI:** 10.3390/md22050217

**Published:** 2024-05-13

**Authors:** Ye-Ji Kim, Yejin Jo, Seung Eun Lee, Jungeun Kim, Jae-Pil Choi, Nayoung Lee, Hyokyoung Won, Dong Ho Woo, Seungshic Yum

**Affiliations:** 1Department of Advanced Toxicology Research, Korea Institute of Toxicology (KIT), Daejeon 34114, Republic of Korea; yeji.kim@kitox.re.kr; 2Human and Environmental Toxicology, University of Science and Technology, Daejeon 34114, Republic of Korea; 3Ecological Risk Research Division, Korea Institute of Ocean Science and Technology (KIOST), Geoje 53201, Republic of Korea; ye9302@kiost.ac.kr (Y.J.); dylee@kiost.ac.kr (N.L.); bluepigret@naver.com (H.W.); 4Research Animal Resource Center, Korea Institute of Science and Technology (KIST), Seoul 02792, Republic of Korea; selee@kist.re.kr; 5Personal Genomics Institute (PGI), Genome Research Foundation (GRF), Cheongju 28160, Republic of Korea; jungeunkim079@gmail.com (J.K.); casperch@gmail.com (J.-P.C.); 6KIOST School, University of Science and Technology, Geoje 53201, Republic of Korea

**Keywords:** toxin, venom, jellyfish, Cnidaria, genomic information, electrophysiology

## Abstract

We identified a new human voltage-gated potassium channel blocker, NnK-1, in the jellyfish *Nemopilema nomurai* based on its genomic information. The gene sequence encoding NnK-1 contains 5408 base pairs, with five introns and six exons. The coding sequence of the NnK-1 precursor is 894 nucleotides long and encodes 297 amino acids containing five presumptive ShK-like peptides. An electrophysiological assay demonstrated that the fifth peptide, NnK-1, which was chemically synthesized, is an effective blocker of hKv1.3, hKv1.4, and hKv1.5. Multiple-sequence alignment with cnidarian Shk-like peptides, which have Kv1.3-blocking activity, revealed that three residues (^3^Asp, ^25^Lys, and ^34^Thr) of NnK-1, together with six cysteine residues, were conserved. Therefore, we hypothesized that these three residues are crucial for the binding of the toxin to voltage-gated potassium channels. This notion was confirmed by an electrophysiological assay with a synthetic peptide (NnK-1 mu) where these three peptides were substituted with ^3^Glu, ^25^Arg, and ^34^Met. In conclusion, we successfully identified and characterized a new voltage-gated potassium channel blocker in jellyfish that interacts with three different voltage-gated potassium channels. A peptide that interacts with multiple voltage-gated potassium channels has many therapeutic applications in various physiological and pathophysiological contexts.

## 1. Introduction

Technological advances in genome and transcriptome sequencing, bioinformatics, and the chemical synthesis of peptides and proteins provide unprecedented opportunities to isolate potential pharmacomedical compounds from venoms. Cnidaria is a representative group of venomous marine animals. Various kinds of potentially toxic proteins and peptides have been reported in cnidarians after transcriptomic and/or proteomic analyses, and they have been reviewed recently [1,2,3,4]. The functional assays of some toxic components deduced from transcriptomic analyses have been undertaken with chemically synthesized materials rather than purified toxic components [5]. *Nemopilema nomurai* venom has great potential utility in this regard because its genomic information is available [6], and its venom extracts have shown various therapeutic properties, including antimetastatic [7] and anticancer effects [8]. Therefore, it is highly likely that jellyfish venom contains many types of ion channel blockers.

Voltage-gated potassium channels play critical roles in regulating the electrical activity of cells, particularly in neurons and muscle cells [9]. Investigating the binding specificity and selectivity of toxins to different potassium channel subtypes may provide insights into the structural determinants of channel–peptide interactions and aid in the design of more selective modulators or drugs targeting specific potassium channel isoforms. A comprehensive review of Kv1 channels targeting toxins from marine animals is available [10]. Among Kv1 channels, the activation of the voltage-gated potassium channel Kv1.3 in human T and B lymphocytes is related to the development of autoimmune diseases [11,12,13]. Therefore, Kv1.3 has become a therapeutic target for the treatment of these diseases [14,15,16].

Among the effective potassium channel blockers, an ShK-186 analog was the first candidate drug to display clinically useful traits [17,18]. The ShK peptide was identified in the sea anemone *Stichodactyla helianthus* [19]. In brief, it consists of 35 amino acid residues, including six cysteines bridged by three disulfide bonds [20], and it blocks voltage-dependent potassium channels. Since its discovery, other toxic ShK-like peptides have been detected and characterized in other sea anemone species [21,22,23,24,25] and corals [5,26]. ShK-like peptides have also been reported in jellyfish species [27,28]; however, their biological function has not yet been revealed.

Recently, we identified seven ShK-like peptide precursor genes in the genomic information of the jellyfish *N*. *nomurai* [6]. In the present study, we describe the structure of an open reading frame (ORF) among these seven precursor genes and its deduced amino acid sequence, which contains five putative ShK-like peptides. The fifth peptide, NnK-1, was chemically synthesized, and we investigated whether it has voltage-gated-potassium-channel-blocking activity in the isoforms and subtypes of human potassium channels. For this purpose, we used an internal ribosome entry site (IRES)-containing vector to separately express the human potassium channel genes (*hKv1.1*, *hKv1.3*, *hKv1.4*, *hKv1.5*, *hKv3.1*, and *hKv11.1*) and the enhanced green fluorescent protein (EGFP) gene to avoid generating an EGFP–Kv fusion protein. The current amplitudes at the voltage increments of +50 mV were significantly reduced in cells expressing synthetic NnK-1. Our findings contribute to the characterization of toxins with potential voltage-gated-ion-channel-blocking functions, which could be developed for therapeutic applications.

## 2. Results and Discussion

### 2.1. Genomic DNA and Transcript Sequences of NnK-1 Precursor Gene

The whole gene sequence of the NnK-1 precursor contains 5408 base pairs with six distinct exons, and the classical 5′ donor (GT) and 3′ acceptor (AG) splice sites are present at each exon–intron boundary (Figure 1).

A transcript encoding an ShK-like peptide was predicted. Figure 2 shows the sequence structure of 894 base pairs encoding the 297 amino acid residues of an ORF. Five presumptive ShK-like peptides (printed in bold) were detected in the protein. All five peptides had mono- or dibasic amino acid residues upstream of their N-termini and downstream of their C-termini, and these were responsible for peptide precursor conversion [29]. Multiple ShK-like peptides are also tandemly arranged in a single coding sequence in the genomes of jellyfish and sea anemones [30]. Therefore, a more efficient process for producing cysteine-rich peptide toxins may have evolved at some point during the evolutionary history of cnidarians, possibly through gene duplication.

### 2.2. Voltage-Gated Potassium Channel Blockade Function of NnK-1

We constructed six kinds of pAAV–CMV–hKv–IRES2–EGFP, which included each of the *hKv1.1*, *hKv1.3*, *hKv1.4*, *hKv1.5*, *hKv3.1*, and *hKv11.1* genes, to allow the activity of these potassium channels to be measured separately without interference by green fluorescent protein (GFP). We mainly focused on measuring the activity of hKv1.3. After the transfection of pAAV–CMV–hKv1.3–IRES2–EGFP, the fluorescence of GFP was highly visible (Figure 3A), and the currents from the HEK293 cells expressing AAV–CMV–hKv1.3–IRES2–EGFP were clearly recorded when the specified voltage steps were applied, unlike in the untransfected HEK293 cells (Figure 3B,C). The current–voltage (IV) recorded in the HEK cells expressing hKv1.3 showed a significant difference in the interaction between the expressed hKv1.3 and the voltage steps (Figure 3C). The hKv1.3 current amplitudes at 50 mV were significantly higher than those of the untransfected HEK293 cells (Figure 3D). The current of hKv1.3 was significantly reduced by treatment with 0.01, 1, 100, or 500 nM synthetic ShK peptide, which was first identified in the sea anemone *S*. *helianthus* [19] (Figure 3E–G, * *p* < 0.05, ** *p* < 0.01, *** *p* < 0.001, **** *p* < 0.0001), as well as by treatment with 0.01, 1, 100, or 500 nM Psora4, a pan-Kv1.3 blocker (Figure 3H–J, * *p* < 0.05, ** *p* < 0.01, *** *p* < 0.001, **** *p* < 0.0001). Importantly, the current amplitude of hKv1.3 was also significantly reduced through the application of 0.01, 1, 100, or 500 nM NnK-1 (Figure 3K–M, * *p* < 0.05, ** *p* < 0.01, **** *p* < 0.0001), suggesting that NnK-1 is a candidate inhibitor of hKv1.3. To assess whether the toxicity of ShK and NnK-1 affected their inhibitory effect on hKv1.3, we measured the cell viability in the untransfected HEK293 cells and hKv1.3-transfected HEK293 cells. There was no difference in 1 h treatments with 0.01, 1, 100, and 500 nM ShK and NnK-1 in the untransfected HEK293 cells and hKv1.3-transfected HEK293 cells (Figure 3N–Q).

We also found that NnK-1 (100 nM) significantly inhibited the current amplitude of hKv1.3, hKv1.4, and hKv1.5, but not that of hKv1.1, hKv3.1, and hKv.11.1 (Figure 4, * *p* < 0.05, ** *p* < 0.01, *** *p* < 0.001). These results implied that hKv1.3, hKv1.4, and hKv1.5 had a similar structure. The structure of hKv1.3, hKv1.4, and hKv1.5 has been recently studied [31]. These channels exhibit a similar overall structure, characterized by six transmembrane segments (S1–S6), with the S4 segment serving as the voltage sensor and the pore-forming region located between S5 and S6. Kv1.3 consists of four alpha subunits, each containing the six transmembrane segments. The S4 segment functions as the voltage sensor, housing positively charged amino acids that respond to changes in membrane potential. The pore region, formed by the S5 and S6 segments, facilitates the passage of potassium ions through the membrane. Similarly, Kv1.4 comprises four alpha subunits with six transmembrane segments. Kv1.5 channels also share the general structure of the Kv1 family channels, featuring six transmembrane segments per alpha subunit. It is intriguing that these three isoforms share structural similarities despite being expressed in different cell types or tissues. For instance, hKv1.3 channels are widely expressed in various tissues, including immune cells (such as T lymphocytes), neurons, and other cell types [12,32]; hKv1.4 channels are widely expressed in various tissues, including the nervous system, skeletal muscle, and heart [33,34]; while hKv1.5 channels are predominantly expressed in the heart [35,36].

### 2.3. Structural Similarity between Sea Anemone ShKs and Jellyfish NnK-1

The amino acid sequences of the voltage-gated potassium channel blockers identified in sea anemones (ShK, BgK, HmK, AeK, AsKs, OsPTx2a, and AEtxK1) and that in the jellyfish (NnK-1) were compared (Figure 5). The aspartic acid (D) in the third position of the peptide is conserved in the ShK analogs from both sea anemones and jellyfish. The one-amino-acid residues (T) located just before the fifth cysteine were also conserved in all the sequences. However, the positions of the two potential key binding residues (KY) were conserved in the sea anemone peptides but not in the jellyfish peptide. Therefore, three residues (^3^Asp, ^25^Lys, and ^34^Thr in NnK-1) were likely to be essential for the binding of the toxin to the voltage-gated potassium channels. A peptide with these three amino acids, substituted with ^3^Glu, ^25^Arg, and ^34^Met, named NnK-1 mu, was synthesized. Then, an electrophysiological assay was carried out with this peptide to confirm whether these three amino acid residues had a crucial role in the binding of the toxins to the voltage-gated potassium channels.

Another peptide, NnK-1 w/o d, which had the same amino acid residues but without three pairs of disulfide bonds, was also synthesized to test whether the disulfide bonds were still intact in the synthetic NnK-1 after being dissolved in a solution. The results showed that neither NnK-1 mu nor NnK-1 w/o d inhibited the current amplitudes of hKv1.3 (Figure 6, * *p* < 0.05). Thus, the three residues and three pairs of disulfide bonds were necessary for potassium-channel-blocking activity, and the disulfide bonds in synthetic NnK-1 were maintained even when dissolved in a solution.

Meanwhile, the Shk-like peptides PcShK3 and AmAMP1 have been identified in corals. PcShK3 exerts both neuro- and cardioprotective effects in zebrafish [5], and AmAMP1 has antimicrobial activity [26]. Moreover, peptides with a cysteine-rich ShK motif that are expressed in neurons have been identified in *Nematostella vectensis*, a sea anemone model organism, together with a peptide expressed in nematocysts [37]. The position of aspartic acid (D) is conserved in ShK peptides, and it has both toxic and other functions (Figure 7). However, the KT motif is only conserved in ShK peptides classified as toxins (Figure 7).

In conclusion, we successfully identified and characterized a new voltage-gated potassium channel blocker in jellyfish, NnK-1, which interacts with three different voltage-gated potassium channels (hKv1.3, hKv 1.4, and hKv1.5). The discovery of a peptide that interacts with multiple voltage-gated potassium channels has significant implications for our knowledge of ion channel function, drug discovery, and potential therapeutic applications in various physiological and pathophysiological contexts. We also showed that using synthetic peptides based on genomic information, rather than purified peptides, is an efficient way to identify the biomaterials that can be developed for therapeutic application in the treatment of human diseases.

## 3. Materials and Methods

### 3.1. In Silico Identification of ShK-Like Peptide Genes in N. nomurai

The ShK domains (PF01549.26) of *N*. *nomurai* genes were identified with the Protein Families database (Pfam ver. 34.0). We confirmed the number of cysteine residues in the ShK domains and the presence of basic residues within the 10 amino acids flanking the ShK domains. Finally, we manually selected the ShK-like peptides to be synthesized.

### 3.2. Peptide Synthesis

NnK-1, NnK-1 mu, and NnK-1 w/o d (Table 1) were synthesized with a standard solid-phase methodology, followed by trifluoroacetic acid–anisole (81.5% TFA + 2.5% EDT + 5% H_2_O + 5% phenol + 5% thioanisole + 1% TIS) cleavage and high-performance liquid chromatographic purification (Appendix A). Three disulfide bridges formed between Cys1 and Cys38, Cys11 and Cys31, and Cys20 and Cys35 but not in NnK-1 w/o d. The molecular weight of the synthetic peptide was confirmed with matrix-assisted laser desorption–ionization–time-of-flight (MALDI-TOF)–mass spectrometry (MS) (Shimadzu LCMS-2020, Kyoto, Japan) (Appendix A). The peptide was synthesized by Pepmic Co. Ltd. (Suzhou, China).

### 3.3. Cloning hKv cDNAs and Construction of hKv Expression Vectors

The primer sets used to amplify the hKvs (*hKv1.1*, *hKv1.3*, *hKv1.4*, *hKv1.5*, *hKv3.1*, and *hKv11.1*) gene are provided in Table 2. The amplification was performed in the Veriti™ 96-Well Fast Thermal Cycler (Applied Biosystems, Waltham, MA, USA) with a thermal cycling program consisting of predenaturation for 3 min at 95 °C, followed by 30 cycles of denaturation for 30 s at 95 °C, annealing for 30 s at 58 °C, and extension for 30 s at 72 °C, with a final extension for 3 min at 72 °C. A Human cDNA Clone Set (OriGene Technologies GmbH, Herford, Germany) was used as the template. A MCS (multi-cloning site) digested from the pAAV-MCS vector (Cell Biolabs, San Diego, CA, USA) was ligated into the pIRES2-EGFP vector (Clontech Laboratories, Mountain View, CA, USA) to form pAAV–CMV–IRES2–EGFP. The PCR products were ligated into the pAAV–CMV–IRES2–EGFP vector at the NheI and SalI sites to construct pAAV–CMV–hKv–IRES2–EGFP.

### 3.4. Preparation of hKv-Vector-Transformed HEK293 Cells

The HEK293 cells were transfected with pAAV–CMV–hKv–IRES2–EGFP (2 μg) using Effectene Transfection Reagent (Qiagen, Hilden, Germany) for 12–18 h. The transfectants (1 × 10^5^/mL) were seeded onto 12 mm coverslips coated with 0.01 mg/mL poly-D-lysine for 2–3 h.

### 3.5. Current Recording

The HEK293 cells expressing AAV–CMV–hKv–IRES2–EGFP were reseeded onto coverslips. The HEK293 transfectants were monitored for their green fluorescence and used as patch cells. Patch pipettes were filled with an internal solution (140 mM potassium gluconate, 10 mM HEPES, 0.2 mM ATP, and 0.06 mM GTP). An external solution [130 mM NaCl, 10 mM HEPES, 3 mM KCl, 1.5 mM D-glucose, 10 mM sucrose, 24 mM CaCl_2_, 1.5 mM MgCl_2_·6H_2_O, osmolarity 320 mmol/kg (pH 7.2)] was used as the basic buffer. Psora4, ShK, or NnK-1 (0, 0.01, 1, 100, or 500 nM) was bath-applied for 2 min to block the currents recorded from the HEK293 cells expressing AAV–CMV–hKv–IRES2–EGFP with voltages ranging from −130 to 50 mV in the increments of 20 mV and 0.6 s. Whole-cell patch recordings from cultured cortical neurons under voltage-clamp conditions (holding potential of –70 mV) were made with the Multiclamp 700 B microelectrode amplifier (Automate Scientific Inc., Berkeley, CA, USA) digitized with a Digidata 1322 A data acquisition system (Molecular Devices Limited, San Jose, CA, USA). In this study, all electrophysiological data from the cultured cells were obtained at a temperature of 23–25 °C and maintained with the CL-100 Temperature Controller (Warner Instruments LLC, Hamden, CT, USA).

### 3.6. Cell Viability Assay

The cell viability of the HEK293 cells was assessed with a Cell Counting Kit-8 (CCK-8, DJDB4000X, Dojindo laboratory, Kumanoto, Japan). Both the HEK293 cells and HEK293 cells expressing AAV–CMV–hKv–IRES2–EGFP were seeded onto a 96-well plate (1 × 10^4^ cells/well). After 24 h of incubation, ShK, NnK-1, NnK-1 mu, and NnK-1 w/o d (0, 0.01, 1, 100, or 500 nM) were used to treat the cells in each well for 1 h. After that, a dye solution was used to treat the cells for 2 h, and the absorbance was measured at 450 nm using a microplate reader (GloMax explorer multimode, Promega, Madison, WI, USA).

## Figures and Tables

**Figure 1 marinedrugs-22-00217-f001:**
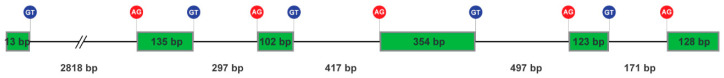
Organization of the *Nemopilema nomurai* NnK-1 precursor gene, which contains six distinct exons (green).

**Figure 2 marinedrugs-22-00217-f002:**
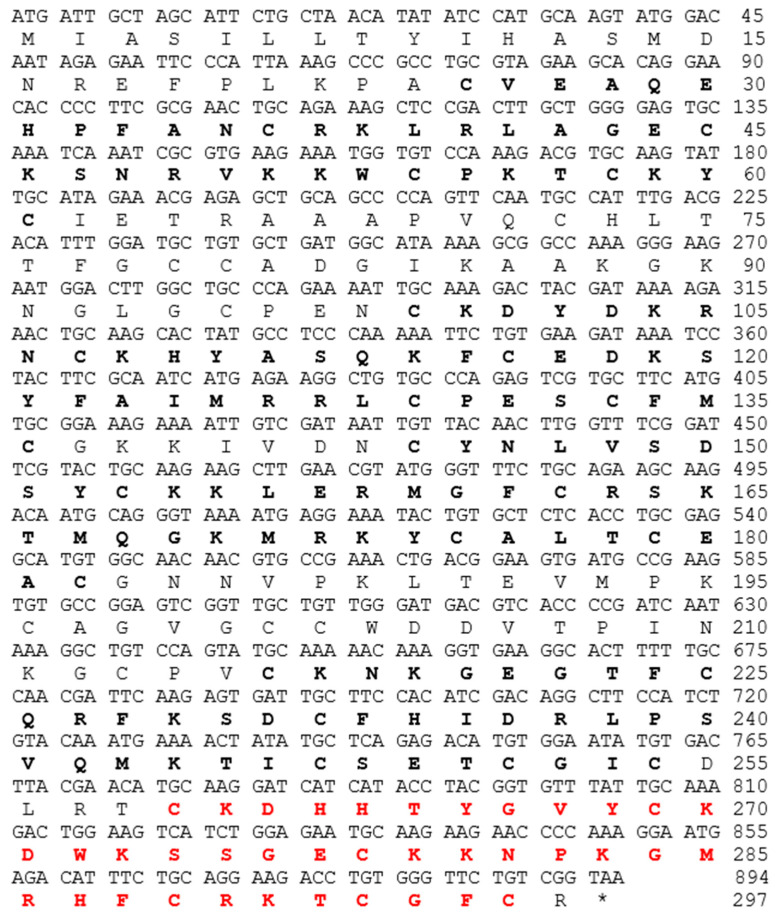
Transcript sequence of the NnK-1-encoding gene and the deduced amino acid sequences. The amino acid sequences of the five ShK-like peptides are printed in bold. NnK-1 is printed in red.

**Figure 3 marinedrugs-22-00217-f003:**
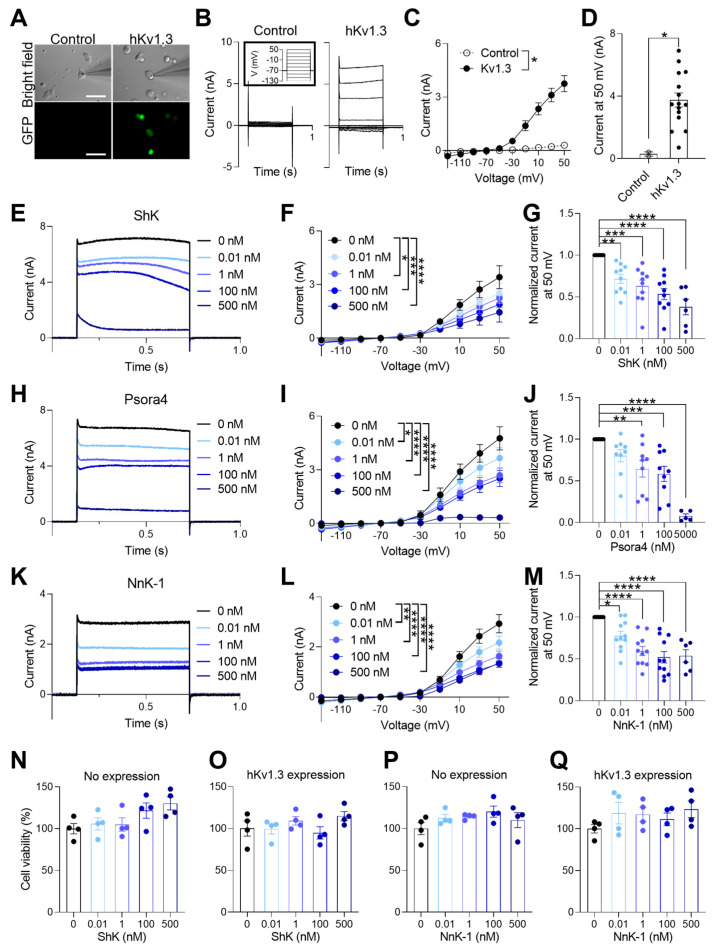
Inhibitory effect of NnK-1 on the activation of hKv1.3 channels. (**A**) Bright-field images of glass pipettes used for the patch clamp of the HEK293 cells (upper panels) and fluorescence microscopic images of the cells (lower panel) without (left panel) and with GFP (right panel) showing the expression of AAV–CMV–hKv1.3–IRES2–EGFP. Scale bar, 50 µm. (**B**) Current amplitudes ranging from −130 to 50 mV in the increments of 20 mV (10 steps) without (left) and with hKv1.3 (right). The inset indicates 10 voltage steps. (**C**) Representative current–voltage (IV) curves without (○) and with hKv1.3 (●). Unpaired *t*-test, * *p* < 0.05. (**D**) The summary bar graphs of current amplitudes without and with hKv.1.3 expression at 50 mV. Unpaired *t*-test, * *p* < 0.05. (**E**) hKv1.3-mediated currents at 50 mV in the presence of the indicated concentrations of ShK (0, 0.01, 1, 100, or 500 nM). (**F**) Representative IV curves in the presence of ShK (0, 0.01, 1, 100, or 500 nM). Two-way ANOVA, interaction between the voltage and group F(36, 410) = 1.269, *p* = 0.14; voltage F(9, 410) = 65.13, *p* < 0.0001; group F(4, 410) = 6.148, *p* < 0.0001, followed by Dunnett’s post hoc test, * *p* < 0.05, 0 vs. 1 nM, *** *p* < 0.001, 0 vs. 100 nM, **** *p* < 0.0001, 0 vs. 500 nM. (**G**) The summary bar graphs of current amplitudes at 50 mV in the presence of ShK (0, 0.01, 1, 100, or 500 nM). One-way ANOVA, F(4, 41) = 13.51, *p* < 0.0001, followed by Dunnett’s post hoc test, ** *p* < 0.01, 0 vs. 0.01 nM, *** *p* < 0.001, 0 vs. 1 nM, **** *p* < 0.0001, 0 vs. 100 nM, 0 vs. 500 nM. (**H**) hKv1.3-mediated currents at 50 mV in the presence of the indicated concentrations of Psora4 (0, 0.01, 1, 100, or 500 nM). (**I**) Representative IV curves in the presence of Psora4 (0, 0.01, 1, 100, or 500 nM). Two-way ANOVA, interaction between the voltage and group F(36, 360) = 4.805, *p* < 0.0001; voltage F(9, 360) = 88.19, *p* < 0.0001; group F(4, 360) = 25.38, *p* < 0.0001, followed by Dunnett’s post hoc test, * *p* < 0.05, 0 vs. 0.01 nM, **** *p* < 0.0001, 0 vs. 1 nM, 0 vs. 100 nM, 0 vs. 500 nM. (**J**) The summary bar graphs of current amplitudes at 50 mV in the presence of Psora4 (0, 0.01, 1, 100, or 500 nM). One-way ANOVA, F(4, 36) = 15.58, *p* < 0.0001, followed by Dunnett’s post hoc test, ** *p* < 0.01, 0 vs. 1 nM, *** *p* < 0.001, 0 vs. 100 nM, **** *p* < 0.0001, 0 vs. 500 nM. (**K**) Representative traces in the presence of the indicated concentrations of NnK-1 (0, 0.01, 1, 100, or 500 nM). (**L**) Representative IV curves in the presence of NnK-1 (0, 0.01, 1, 100, or 500 nM). Two-way ANOVA, the interaction between the voltage and group F(36, 450) = 5.257, *p* < 0.0001; voltage F(9, 450) = 183.1, *p* < 0.0001; group F(4, 450) = 21.75, *p* < 0.0001, followed by Dunnett’s post hoc test, ** *p* < 0.01, 0 vs. 0.01 nM, **** *p* < 0.0001, 0 vs. 1 nM, 0 vs. 100 nM, 0 vs. 500 nM. (**M**) The summary bar graphs of current amplitudes at 50 mV in the presence of NnK-1 (0, 0.01, 1, 100, or 500 nM). One-way ANOVA, F(4, 45) = 14.39, *p* < 0.0001, followed by Dunnett’s post hoc test, * *p* < 0.05, 0 vs. 0.01 nM, **** *p* < 0.0001, 0 vs. 1 nM, 0 vs. 100 nM, 0 vs. 500 nM. (**N**,**O**) The summary bar graph of the HEK293 cells’ viability without hKv1.3 expression (**N**) and with hKv1.3 expression (**O**) under 1 h of ShK (0, 0.01, 1, 100, or 500 nM) treatment. (**O**–**Q**) The summary bar graph of the HEK293 cells’ viability without hKv1.3 expression (**P**) and with hKv1.3 expression (**Q**) under 1 h of NnK-1 (0, 0.01, 1, 100, or 500 nM) treatment. Data are the means ± SEM.

**Figure 4 marinedrugs-22-00217-f004:**
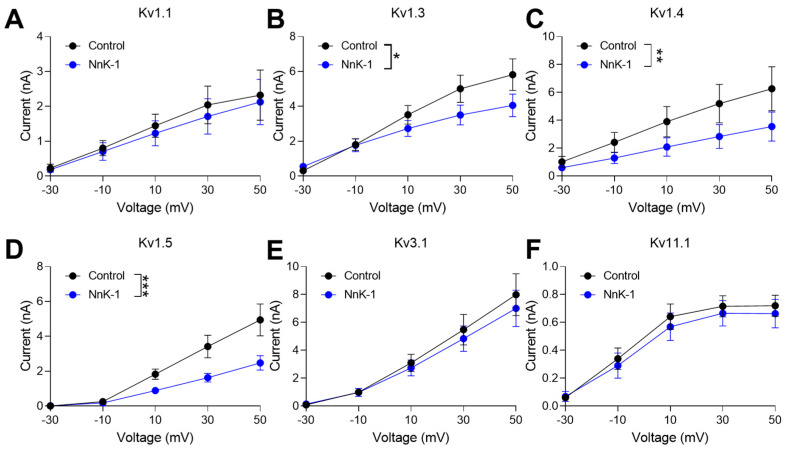
Inhibitory effect of NnK-1 on the activation of hKv channels. (**A**) Representative IV curves in the presence of 100 nM NnK-1 with respect to hKv1.1. Two-way ANOVA, interaction between the voltage and group F(4, 20) = 0.03, *p* = 0.99; voltage F(4, 20) = 7.184, *p* < 0.001; group F(1, 20) = 0.43, *p* = 0.52. (**B**) Representative IV curves in the presence of 100 nM NnK-1 with respect to hKv1.3. Two-way ANOVA, interaction between the voltage and group F(4, 70) = 1.32, *p* = 0.27; voltage F(4, 70) = 22.81, *p* < 0.0001; group F(1, 70) = 5.03, * *p* < 0.05. (**C**) Representative IV curves in the presence of 100 nM NnK-1 with respect to hKv1.4. Two-way ANOVA, interaction between the voltage and group F(4, 30) = 0.49, *p* = 0.74; voltage F(4, 30) = 6.18, *p* < 0.001; group F(1, 30) = 8.17, ** *p* < 0.01. (**D**) Representative IV curves in the presence of 100 nM NnK-1 with respect to hKv1.5. Two-way ANOVA, interaction between the voltage and group F(4, 30) = 3.65, *p* < 0.05; voltage F(4, 30) = 30.85, *p* < 0.0001; group F(1, 30) = 17.51, *** *p* < 0.001. (**E**) Representative IV curves in the presence of 100 nM NnK-1 with respect to hKv3.1. Two-way ANOVA, interaction between the voltage and group F(4, 40) = 0.14, *p* = 0.97; voltage F(4, 40) = 26.93, *p* < 0.0001; group F(1, 40) = 0.56, *p* = 0.46. (**F**) Representative IV curves in the presence of 100 nM NnK-1 with respect to hKv11.1. Two-way ANOVA, interaction between the voltage and group F(4, 40) = 0.07, *p* = 0.99; voltage F(4, 40) = 23.68, *p* < 0.0001; group F(1, 40) = 0.75, *p* = 0.39. Data are the means ± SEM.

**Figure 5 marinedrugs-22-00217-f005:**
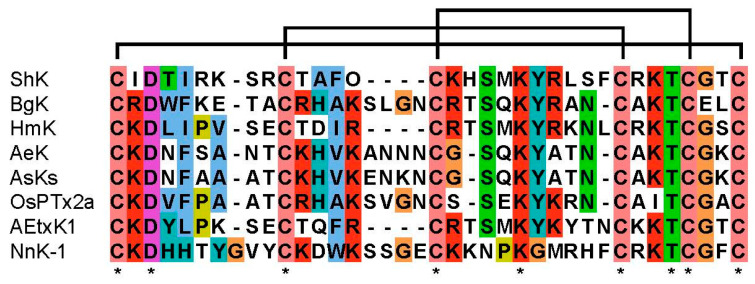
Multiple-sequence alignment of six sea anemone peptides and one jellyfish peptide, all of which show Kv1.3-blocking activity. ShK (from *Stichodactyla helianthus*), BgK (from *Bunodosoma granulifera*), HmK (from *Heteractis magnifica*), Aek (from *Actinia equina*), AsKs (from *Anemonia sulcata*), OsPtx2a (from *Oulactis* sp.), AEtxK1 (from *Anemonia erythraea*), and NnK-1 (from *N. nomurai*). Conserved amino acid residue marked with an asterisk.

**Figure 6 marinedrugs-22-00217-f006:**
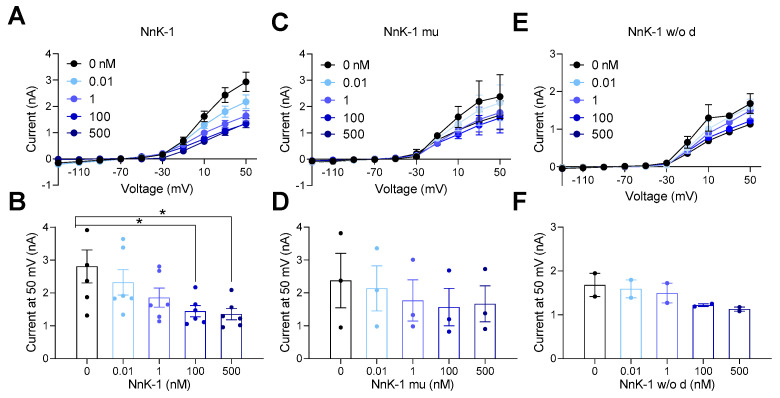
Inhibitory effects of NnK-1 analogs on the activation of hKv1.3 channels. (**A**) Representative IV curves in the presence of NnK-1 (0, 0.01, 1, 100, or 500 nM). (**B**) The summary bar graphs of current amplitudes at 50 mV in the presence of NnK-1 (0, 0.01, 1, 100, or 500 nM). One-way ANOVA, F(4, 25) = 3.42, *p* < 0.05, followed by Dunnett’s post hoc test, * *p* < 0.05, 0 vs. 100 nM, 0 vs. 500 nM. (**C**) Representative IV curves in the presence of NnK-1 mu (0, 0.01, 1, 100, or 500 nM). (**D**) The summary bar graphs of current amplitudes at 50 mV in the presence of NnK-1 mu (0, 0.01, 1, 100, or 500 nM). One-way ANOVA, F(4, 10) = 0.27, *p* = 0.89. (**E**) Representative IV curves in the presence of NnK-1 w/o d (0, 0.01, 1, 100, or 500 nM). (**F**) The summary bar graphs of current amplitudes at 50 mV in the presence of NnK-1 w/o d (0, 0.01, 1, 100, or 500 nM). One-way ANOVA, F(4, 5) = 1.71, *p* = 0.28. Data are the means ± SEM.

**Figure 7 marinedrugs-22-00217-f007:**
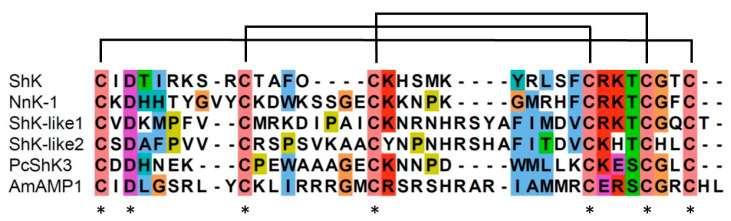
Multiple-sequence alignment of six ShK peptides. ShK, NnK-1, and ShK-like 1 peptides (from *Nematostella vectecsis*) are known to be toxins. ShK-like 2 (from *N*. *vectecsis*) is expressed in neurons, PcShK3 (from *Palythoa cariboeorum*) has neuro- and cardioprotective functions, and AmAMP1 (from *Acropora millepora*) has antimicrobial activity. Conserved amino acid residue marked with an asterisk.

**Table 1 marinedrugs-22-00217-t001:** The primary structure of the synthetic peptides used in the assays.

Name	Sequence	Disulfide Bonds
NnK-1	CKDHHTYGVYCKDWKSSGECKKNPKGMRHFCRKTCGFC	O
NnK-1-mu	CKEHHTYGVYCKDWKSSGECKKNPRGMRHFCRKMCGFC	O
NnK-1-w/o d	CKDHHTYGVYCKDWKSSGECKKNPKGMRHFCRKTCGFC	X

**Table 2 marinedrugs-22-00217-t002:** The primer sets used to amplify the hKv channel genes with polymerase chain reaction.

Gene	Primer	Sequence
*hKv1.3*	NheI-Kv1.3-F Sal1-Kv1.3-R	5′-TTTGCTAGCGCCACCATGGACGAGCGC-3′ 5′-TTTGTCGACCTAAACATCGGTGAATATCTTTT-3′
*hKv1.1*	NheI-Kv1.1-F Sal1-Kv1.1-R	5′-AACCGTCAGATCCGCTAGCGCCACCATGACGGTGATGTCTGGG-3′ 5′-GAGGGGCGGTACCGTCGACTTAAACATCGGTCAGTAGC-3′
*hKv1.4*	NheI-Kv1.4-F Sal1-Kv1.4-R	5′-TGAACCGTCAGATCCGCTAGCGCCACCATGGAGGTTGCAATGGTG-3′ 5′-GAGAGGGGCGGTACCGTCGACTCACACATCAGTCTCCAC-3′
*hKv1.5*	NheI-Kv1.5-F Sal1-Kv1.5-R	5′-AACCGTCAGATCCGCTAGCGCCACCATGGAGATCGCCCTGGTG-3′ 5′-GAGGGGCGGTACCGTCGACTCACAAATCTGTTTCCCG-3′
*hKv3.1*	NheI-Kv3.1-F Sal1-Kv3.1-R	5′- AACCGTCAGATCCGCTAGCGCCACCATGGGCCAAGGGGACGAG-3′ 5′-GAGGGGCGGTACCGTCGACTCAAGTCACTCTCACAGC-3′
*hKv11.1*	NheI-Kv11.1-F Sal1-Kv11.1-R	5′-AGTGAACCGTCAGATCCGCTAGCGCCACCATGCCGGTGCGGAGGGGC-3′ 5′-GGGAGAGGGGCGGTACCGTCGACCTAACTGCCCGGGTCCGAG-3′

## Data Availability

The data that support the findings of this study are available in the figures of the article.

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
