# Peer review of "Synthetic ShK-like Peptide from the Jellyfish Nemopilema nomurai Has Human Voltage-Gated Potassium-Channel-Blocking Activity"

_marinedrugs, 2024, doi:10.3390/md22050217_

Round 1

Reviewer 1 Report (Previous Reviewer 1)

Comments and Suggestions for Authors

In this manuscript, the authors summarized and discussed ShK-like peptide NnK-1 from the jellyfish Nemopilema nomurai may be a new blocker for human voltage-gated potassium-channel. On this basis, they further investigated structural similarity between sea anemone ShKs and jellyfish NnK-1, and the blocking effect after changing the conservative sites and disulfide bonds. The manuscript is well-organized and clearly stated. I would suggest accepting it after the following minor concerns are addressed.

1.      If there is direct evidence to prove that PAAV-CMV-hKv IRES2-EGFP is transfected and expressed and localized on the HEK293 plasma membrane, the experiment may be more complete.

2.      Please explain why two positive controls, Shk and Psora4, were chosen instead of using only Shk or Psora4 alone. Additionally, the order of Shk (E-G) and Psora4 (H-J) in Fig.3 can be swapped.

3.      After line 160, please show the structure of hKv1.3, hKv1.4 and hKv1.5 and explain their structural similarity in detail.

4.      Please explain why the mutant amino acids in NnK-1 mu are 3Glu, 25Arg and 34Met, but not other amino acids.

5.      If the mechanism of action of three amino acid sites and disulfide bonds is supplemented to elucidate the molecular mechanism of NnK-1 blocking voltage-gated potassium channels, the entire experiment will be very complete.

6.      Please explain why the current of the NnK-1 group is higher than that of the control group when the voltage in Fig. 4B is -30mV.

7.      The abstract section and keywords need to be improved. Suggest adding keywords related to Shk-like Peptide and voltage-gated potassium channel.

Author Response

  1. If there is direct evidence to prove that PAAV-CMV-hKv IRES2-EGFP is transfected and expressed and localized on the HEK293 plasma membrane, the experiment may be more complete.

ANS) The reviewer suggested that this manuscript would be further refined by providing information on whether the AAV-CMV-hKv IRES2-EGFP clone is expressed on the cell membrane, and we agree with the reviewer's opinion. Nevertheless, we regret that we cannot carry out this experiment at the moment. We believe that the evidence we presented is sufficient to show that our peptide inhibits the channel. We presented pictures of fluorescence, measured activity through electrophysiology, and demonstrated the expression and function of the AAV-CMV-hKv IRES2-EGFP clone expressed in Hek cells through various positive control experiments (Fig. 3). We believe that such information is outside of the flow of this manuscript.

  1. Please explain why two positive controls, Shk and Psora4, were chosen instead of using only Shk or Psora4 alone. Additionally, the order of Shk (E-G) and Psora4 (H-J) in Fig.3 can be swapped.

ANS) ShK is a peptide with 35 amino acid residues, and Psora-4 is a chemical. Research is still ongoing on how ShK and Psora-4 structurally suppress Kv1.3. Therefore, the use of ShK, a peptide-type inhibitor, and the use of chemical inhibitors were significant in artificially expressing AAV-CMV-hKv IRES2-EGFP in HEK cells and confirming the activity of this channel.

  1. After line 160, please show the structure of hKv1.3, hKv1.4 and hKv1.5 and explain their structural similarity in detail.

ANS) We added some structural information of hKv1.3, hKv1.4 and hKv1.5 after the line 160, as follows.

The structure of hKv1.3, hKv1.4, and hKv1.5 has been recently studied (Rohaim et al., 2022). These channels exhibit a similar overall structure, characterized by six transmembrane segments (S1-S6), with the S4 segment serving as the voltage sensor and the pore-forming region located between S5 and S6. Kv1.3 consists of four alpha subunits, each containing the six transmembrane segments. The S4 segment functions as the voltage sensor, housing positively charged amino acids that respond to changes in membrane potential. The pore region, formed by the S5 and S6 segments, facilitates the passage of potassium ions through the membrane. Similarly, Kv1.4 comprises four alpha subunits with six transmembrane segments. Kv1.5 channels also share the general structure of Kv1 family channels, featuring six transmembrane segments per alpha subunit. It is intriguing that these three isoforms share structural similarities despite being expressed in different cell types or tissues.

  1. Please explain why the mutant amino acids in NnK-1 mu are 3Glu, 25Arg and 34Met, but not other amino acids.

ANS) Multiple-sequence alignment of peptides of six sea anemone and one jellyfish all of which have Kv1.3-blocking activity showed three amino acid residues, 3Asp, 25Lys, and 34Thr, were conserved (Fig. 5). Thus, these three residues were substituted with 3Glu, 25Arg and 34Met, acidic one to acidic (Asp -> Glu; Thr -> Met); basic one to basic (Lys -> Arg) to confirm whether these three amino acid residues had a critical role.  

  1. If the mechanism of action of three amino acid sites and disulfide bonds is supplemented to elucidate the molecular mechanism of NnK-1 blocking voltage-gated potassium channels, the entire experiment will be very complete.

ANS) We think there may be more to the exact mechanism. In order to conduct elaborate mechanistic studies using synthesized peptides, there are other regulatory parts in addition to the length of the peptide and the parts we have pointed out, so it is appropriate to leave this elaborate mechanistic study to the next step. In our research results, a new ShK-like peptide from the giant Nomura's jellyfish inhibits Kv1.3.

  1. Please explain why the current of the NnK-1 group is higher than that of the control group when the voltage in Fig. 4B is -30mV.

The reviewer pointed out the increased current at -30mV with the treatment of 100mM NnK-1, compared to that without the NnK-1. However, the difference of current amplitude at -30mV is quite not significant in statistics. In addition to the tiny difference in statistics, the results of the step current experiment are from the paired experiment and we conclude that NnK-1 reduced the current amplitude at -30mV from the same cell, which means the initial state of the current amplitude at -30mV. Please see blue color dot line of figure 3L showing 100nM NnK-1 did not increase current amplitude at -30mV. There is nothing to discuss as it is not statistically significant.

  1. The abstract section and keywords need to be improved. Suggest adding keywords related to Shk-like Peptide and voltage-gated potassium channel.

The terms which you have recommended were already appeared in the title. Thus, we believe that they are not necessary to be repeated in the keywords.

Reviewer 2 Report (Previous Reviewer 2)

Comments and Suggestions for Authors

No more critism.

Author Response

No more critism.

ANS) Thank you very much for reviewing our manuscript.

Reviewer 3 Report (Previous Reviewer 3)

Comments and Suggestions for Authors

The manuscript has been significantly improved and I am happy to endorse for publishing pending the editors decision. I thank the authors for taking on board the comment to elevate the manuscript. Just double heck the referencing formatting (no. 9) in the proofs once the changes have been accepted.

Author Response

The manuscript has been significantly improved and I am happy to endorse for publishing pending the editors decision. I thank the authors for taking on board the comment to elevate the manuscript. Just double heck the referencing formatting (no. 9) in the proofs once the changes have been accepted.

ANS) Thank you very much for reviewing our revised manuscript.

This manuscript is a resubmission of an earlier submission. The following is a list of the peer review reports and author responses from that submission.

Round 1

Reviewer 1 Report

Comments and Suggestions for Authors

Small peptide potassium ion channel inhibitors usually stabilized the molecular conformation with 2-4 disulfide bonds before exerting their inhibitory effects. The molecular conformation needs to be characterized before and after the dissolution of small peptides.

Some small peptides, especially those derived from venom, are likely to have cytotoxicity, and the inhibitory effect may also be the result of cell damage rather than actual inhibition of potassium ion channels. The core experiment lacked negative and positive controls.

Four key amino acids (3Asp, 25Lys, 33Lys, and 34Thr) required mutations to demonstrate their importance.

Comments on the Quality of English Language

Moderate editing of English language required

Author Response

All the authors really appreciate to the valuable and critical comments of reviewers. We answered to the reviewers and revised the manuscript accordingly as follows.

Reviewer 1

Comments and Suggestions for Authors

  1. Small peptide potassium ion channel inhibitors usually stabilized the molecular conformation with 2-4 disulfide bonds before exerting their inhibitory effects. The molecular conformation needs to be characterized before and after the dissolution of small peptides. 

ANS) Thanks for your concern about the peptide structure before and after dissolved by a solution. NMR spectroscopy is necessary to solve this issue. However, we are not able to carry out the analysis at the moment. The synthetic ShK, peptide which was used as a positive control in our study (Fig. 3), showed hKv1.3 blocking activity. It told us that the three pairs of disulfide bonds in the peptide are likely to not damaged after the dissolution. We believed that the same thing could be happening in the NnK-1 case.

  1. Some small peptides, especially those derived from venom, are likely to have cytotoxicity, and the inhibitory effect may also be the result of cell damage rather than actual inhibition of potassium ion channels. The core experiment lacked negative and positive controls. 

ANS) Thank you for comments. We agree with the reviewer’s comment for NnK-1 toxicity.

We tested cell viability for 1hr with the treatment of ShK and NnK-1 (0, 0.01, 1, 100, 500 nM) on both HEK293 cells and on HEK293 expressing AAV-CMV-hKv1.3-IRES2-EGFP. The durations of Psora-4, ShK, and NnK-1 treatment for the patch clamp are within 1hr. We added Fig. 3N-3Q and NnK-1 did not affect the cell viability.

  1. Four key amino acids (3Asp, 25Lys, 33Lys, and 34Thr) required mutations to demonstrate their importance.

ANS) We will carry out it in our next project with many other candidates. Instead of the analysis, we revised the sentence to tone down the result in line 151-153, as “Therefore, three residues (3Asp, 25Lys, and 34Thr in NnK-1) are likely to be essential for the binding of the toxin to voltage-gated potassium channels.” 33Lys was omitted by a reason.

Reviewer 2 Report

Comments and Suggestions for Authors

Ye-Ji Kim et.al e identified a new human voltage-gated potassium channel (hKv1.3) blocker by analysing the jellyfish genome. NnK-1, an effective hKv1.3 blocker was selected for functional analysis. Multiple sequence alignment suggested that four residues (3Asp, 25Lys, 33Lys, and 34Thr) of NnK-1 are crucial for the binding to the Kv1.3 channels. In summary, the manuscript is well organized. However, there is some over-interpretation.

1.      The authors did not purify the peptide from the jellyfish, and NnK-1 is a putatively deduced peptide. Therefore, the title should modify accordingy.

2.      The gating dynamics of Kv1.3 channels in Figure 3E, 3H and 3K are different, please confirm.

3.      The dose-response of toxins acting on Kv1.3 channels should be fitted with hill equation. However, the results exhibited in Figure 3G, J, and M did not follow this rule.

4.      Although the four residues (3Asp, 25Lys, 33Lys, and 34Thr) are conserved in many functional toxins, there is also essential for the functional test of points mutations of these amino acids. As part of results, there is a lack of evidence.

5.      The method is really simple, it is hard to repeat all the details.

Comments on the Quality of English Language

Moderate editing of English language required

Author Response

All the authors really appreciate to the valuable and critical comments of reviewers. We answered to the reviewers and revised the manuscript accordingly as follows.

Reviewer 2

Comments and Suggestions for Authors

Ye-Ji Kim et.al e identified a new human voltage-gated potassium channel (hKv1.3) blocker by analysing the jellyfish genome. NnK-1, an effective hKv1.3 blocker was selected for functional analysis. Multiple sequence alignment suggested that four residues (3Asp, 25Lys, 33Lys, and 34Thr) of NnK-1 are crucial for the binding to the Kv1.3 channels. In summary, the manuscript is well organized. However, there is some over-interpretation.

  1. The authors did not purify the peptide from the jellyfish, and NnK-1 is a putatively deduced peptide. Therefore, the title should modify accordingy.

ANS) We revised the title as “Synthetic ShK-like peptide from the jellyfish Nemopilema nomurai has human potassium voltage-gated channel-blocking activity.”

  1. The gating dynamics of Kv1.3 channels in Figure 3E, 3H and 3K are different, please confirm.

ANS) Thank you for your kind comment. We did record experiments for the confirmation of the traces of hKv1.3. The traces of hKv1.3 channels are changed in 3E, 3H and 3K.

  1. The dose-response of toxins acting on Kv1.3 channels should be fitted with hill equation. However, the results exhibited in Figure 3G, J, and M did not follow this rule.

ANS) We did record experiment from HEK cells expressing hKv 1.3 and added the current amplitudes at +50mV for Fig. 3G, 3H, and 3M. NnK-1 has a mild inhibition effect on hKv.1.3 compared to Psora4.

  1. Although the four residues (3Asp, 25Lys, 33Lys, and 34Thr) are conserved in many functional toxins, there is also essential for the functional test of points mutations of these amino acids. As part of results, there is a lack of evidence.

ANS) Thank you very much for the comment. We will carry out it in our next project with many other candidates. Instead of the analysis, we revised the sentence to tone down the result in line 151-153, as “Therefore, three residues (3Asp, 25Lys, and 34Thr in NnK-1) are likely to be essential for the binding of the toxin to voltage-gated potassium channels.” 33Lys was omitted by a reason.

  1. The method is really simple, it is hard to repeat all the details.

ANS) We changed and added in the materials and methods.

If you think this version is not enough on your criteria, please let us know which part(s) is(are) too simple to repeat the experiments.

Reviewer 3 Report

Comments and Suggestions for Authors

This manuscript describes the characterisation of an ShK-like peptide from a multi domain in a Jellyfish. Any characterisation of peptides is advantageous in understanding variation in the ShKT domain activity. It is important to centime to characterise peptides which are consistently found through sequence similarity and structure bioinformatics but do not modulate ion channels the same manner.

There are questions and corrections that require addressing prior to acceptance for publication. and rework of some figures is required.

Minor formatting issues such as ensuring all species names are italicised in the Reference list.

Sea anemones also have multiple ShK domains please see (Mitchell et al., 2020) and not only one ShK per ORF as stated in L74.

Where have the sequences been deposited for this peptide domain or if already deposited from a previous study please refer to that deposit reference?

I wonder at the use of aligning OspTx2b as opposed to OspTx2a which actually has Kv1.3 activity, whereas OspTx2b does not. This should be added to or replaced.

The alignment does not contain any other Jellyfish ShKT domain/peptides see (Li et al., 2014) or (Ponce et al., 2016) for inclusion or Aurelia aurita (Moon jellyfish) (Medusa aurita) Q0MWV8 · AURE_AURAU.

Figure 3 is far too crowded and I cannot even see the variation in the cells clearly.

Additional References

Li, R. et al. (2014) ‘Jellyfish venomics and venom gland transcriptomics analysis of Stomolophus meleagris to reveal the toxins associated with sting’, Journal of Proteomics, 106, pp. 17–29. doi: 10.1016/j.jprot.2014.04.011.

Mitchell, M. L. et al. (2020) ‘Tentacle transcriptomes of the speckled anemone (Actiniaria: Actiniidae: Oulactis sp.): Venom-related components and their domain structure’, Marine Biotechnology, 22(2), pp. 207–219. doi: 10.1007/s10126-020-09945-8.

Ponce, D. et al. (2016) ‘Tentacle transcriptome and venom proteome of the Pacific Sea Nettle, Chrysaora fuscescens (Cnidaria: Scyphozoa)’, Toxins, 8(4), p. 102. doi: 10.3390/toxins8040102.

Author Response

All the authors really appreciate to the valuable and critical comments of reviewers. We answered to the reviewers and revised the manuscript accordingly as follows.

Review 3

Comments and Suggestions for Authors

This manuscript describes the characterisation of an ShK-like peptide from a multi domain in a Jellyfish. Any characterisation of peptides is advantageous in understanding variation in the ShKT domain activity. It is important to centime to characterise peptides which are consistently found through sequence similarity and structure bioinformatics but do not modulate ion channels the same manner.

  1. There are questions and corrections that require addressing prior to acceptance for publication. and rework of some figures is required.
  2. Minor formatting issues such as ensuring all species names are italicised in the Reference list.

ANS) Thank you for your kind comments. We changed all species to italicize in the Reference list.

  1. Sea anemones also have multiple ShK domains please see (Mitchell et al., 2020) and not only one ShK per ORF as stated in L74.

ANS) Thanks for the information we didn’t realized. We did check only the ShK-like peptides which have been tested their hKv1.3 blocking activity. We revised the sentences as “Multiple ShK-like peptides are tandemly arranged in a single coding sequence in the jellyfish and the sea anemone [28] as well. Therefore, a more efficient process for producing cysteine-rich peptide toxins may have evolved at some point during the evolutionary history of the cnidariansjellyfish, possibly through gene duplication” in line 76-79.” Mitchell et al., 2020 was cited.

  1. Where have the sequences been deposited for this peptide domain or if already deposited from a previous study please refer to that deposit reference?

ANS) The genome paper of the jellyfish is already cited in the manuscript.

Kim, H. M.; Weber, J. A.; Lee, N.; Park, S. G.; Cho, Y. S.; Bhak, Y.; Lee, N.; Jeon, Y.; Jeon, S.; Luria, V.; Karger, A.; Kirschner, M. W.; Jo, Y. J.; Woo, S.; Shin, K.; Chung, O.; Ryu, J. C.; Yim, H. S.; Lee, J. H.; Edwards, J. S.; Manica, A.; Bhak, J.; Yum, S., The genome of the giant Nomura's jellyfish sheds light on the early evolution of active predation. BMC Biol 2019, 17, (1), 28.

  1. I wonder at the use of aligning OspTx2b as opposed to OspTx2a which actually has Kv1.3 activity, whereas OspTx2b does not. This should be added to or replaced.

ANS) Thanks for pointing out our mistake. We changed OspTx2b to OspTx2a in figure 4.

  1. The alignment does not contain any other Jellyfish ShKT domain/peptides see (Li et al., 2014) or (Ponce et al., 2016) for inclusion or Aurelia aurita (Moon jellyfish) (Medusa aurita)Q0MWV8 · AURE_AURAU.

ANS) We compared the amino acid sequence of ShK-like peptides which their function has been revealed. The sentences in the manuscript were revised as “Since its discovery, other toxic ShK-like peptides have been detected and characterized in other sea anemone species [19-23] and corals [5, 24] ShK-like peptides have also been reported in jellyfish species [25, 26]; however, their biological function has not yet been revealed” in line 50-53. We cited these two references.

  1. Figure 3 is far too crowded and I cannot even see the variation in the cells clearly.

ANS) We changed the Figure3 to be clearer. Fonts are increased and redundant letters are omitted for the simplicity

Round 2

Reviewer 1 Report

Comments and Suggestions for Authors

My comments about this manuscipt did not improve. 

Comments on the Quality of English Language

No

Author Response

My comments about this manuscipt did not improve. 

ANS) Thank you very much for your valuable comments.

Reviewer 2 Report

Comments and Suggestions for Authors

This article has been revised by the author, but it has not been substantially improved. Accept after minor revision

Comments on the Quality of English Language

English language fine. No issues detected

Author Response

This article has been revised by the author, but it has not been substantially improved. Accept after minor revision.

ANS) Thank you very much for your comments.

Reviewer 3 Report

Comments and Suggestions for Authors

Thank you for for clarifying and updating the manuscript, it is more detailed for repeatability with methods. I now recommend for publication.

Comments on the Quality of English Language

I always recommend one last proof read of minor English grammatical corrections required.

Author Response

Thank you for for clarifying and updating the manuscript, it is more detailed for repeatability with methods. I now recommend for publication.

I always recommend one last proof read of minor English grammatical corrections required.

ANS) Thanks a lot for your comments.